# Feasibility and potential significance of prophylactic ablation of the major ascending tributaries in endovenous laser ablation (EVLA) of the great saphenous vein: A case series

**Lars Müller** *, Jens Alm

Department of Vascular Surgery, Dermatologikum Hamburg, Hamburg, Germany

* L.mueller@dermatologikum.de

## Abstract

**Data Availability Statement:** All relevant data are within the paper and its Supporting Information files.

### Background

Recurrent varicosities after endovascular laser ablation (EVLA) of the great saphenous vein (GSV) are frequently due to varicose transformed, initially unsealed major ascending tributaries of the saphenofemoral junction (SFJ). Preventive ablation of these veins, especially the anterior accessory saphenous vein, is discussed as an option, along with flush occlusion of the GSV. However, few related data exist to date.

### Methods

A consecutive case series of 278 EVLA procedures of the GSV for primary varicosis in 213 patients between May and December 2019 was retrospectively reviewed. The ablations were performed with a 1470 nm dual-ring radial laser and always included flush occlusion of the GSV, and concomitant ablation of its highest ascending tributaries by additional cannulation and ablation when this seemed anatomically appropriate. The initial technical success, comprising occlusion of the GSV and its major tributaries, was set as the primary endpoint. Possible determinants were explored using downstream multiple logistic regression analysis.

### Results

The early technical success was 92.8%, with the GSV occluded in 99.6% and the highest ascending SFJ tributary, if present, in 92.4%. Additional ablations of ascending tributaries were performed in 171 cases (61.5%), the latter being associated with success (OR 10.39; 95% CI [3.420–36.15]; p < 0.0001). Presence of anterior as opposed to posterior accessory saphenous vein was another positive predictor (OR 3.959; 95% CI [1.142–13,73]; p = 0.027), while a confluence of the tributary in the immediate proximity to the SFJ had a negative impact (OR 0.2253; 95% CI [0.05456–0.7681]; p = 0.0253). An endothermal heat-induced thrombosis (EHIT) ≥ grade 2 was observed in three cases (1.1%).

**Funding:** Both authors of this study are employed by Dermatologikum Hamburg GmbH. As a non-academic, commercial company, the employer and funder provided support in the form of salaries for authors [L.M., J.A.], but did not have any additional role in the study design, data collection and a nalysis, decision to publish, or preparation of the manuscript. The specific roles of these authors are articulated in the 'author contributions' section.

**Competing interests:** As authors of this study [L. M.,J.A.], we declare that we are employed by Dermatologikum Hamburg GmbH, a nonacademic, commercial company. This does not alter our adherence to PLOS ONE policies on sharing data and materials.

## Conclusions

A co-treatment of the tributaries is feasible and could improve the technical success of EVLA if a prophylactic closure of these veins is desired, especially if their distance to the SFJ is short. Its effect on the recurrence rate needs further research.

## Introduction

Thermal ablation techniques such as endovenous laser ablation (EVLA) and radiofrequency ablation for the treatment of great saphenous vein (GSV) insufficiency are widely accepted as standard options. They are implemented in guidelines of several countries, and their efficacy is supported by a large body of evidence [1–3].

However, recurrences may occasionally occur, progressing from initially nonenlarged ascending tributaries of the GSV. This finding results from recent studies based on observation periods of 5 years [4, 5]. The major ascending tributaries of the GSV, herein also termed side or lateral branches, in particular the anterior and the posterior accessory saphenous vein (AASV, PASV), often join the GSV very close to the saphenofemoral junction (SFJ). Occasionally, these veins drain directly into the SFJ or via common trunks with cranial tributaries such as the superficial epigastric vein or the superficial circumflex iliac vein. They are therefore not necessarily occluded by the thermal energy as part of the ablation [5].

Technical developments of the laser systems used towards radial emitting technology, in which the laser energy instead of shooting forward is directly delivered into the vein wall, allow shorter distances to the deep vein when placing the fiber tip than the 1–3 cm previously usually described [6, 7]. This results in a flush occlusion [8] in which no or only a very small residual stump of the GSV remains, which in turn increases the probability of thermally sealing any tributaries present in the SFJ area. Whether and how often this closure of the ascending tributaries actually succeeds has not yet been systematically explored.

So far there are admittedly no clear scientific data that would indicate an advantage for closure of nonenlarged, competent tributaries during thermal treatment. However, some considerations provide viable arguments. First, the analogous conclusion to surgical high ligation and stripping. Here, there are scientific findings that show a correlation between left residual stumps and undisrupted lateral branches of GSV and the development of recurrence [9, 10]. Second, the above-mentioned long-term studies on EVLA, in which a significant proportion of recurrences were attributed to varicose transformed GSV tributaries that were not abolished during primary treatment [4, 5]. Thirdly, simultaneous laser treatment of the tributaries is being promoted by other users of the technique [11], who like us see advantages with respect to sustainability. In our own center, therefore, experience-based, not evidence-based, closure of GSV tributaries is established as a target for intervention.

The aim of this observational study is therefore to use data from our own treatment routine to investigate whether additional treatment of the uppermost ascending GSV tributaries is feasible and safe, and whether it might impact technical success. In addition, the description of the underlying anatomical constellation will provide possible clues that may serve as a basis for further prospective studies.

## Materials and methods

All consecutive patients who underwent primary EVLA with flush occlusion of the GSV under one operator (LM) at the Dermatologikum Hamburg between May and December 2019

(n = 213) were included and analyzed. For access from April to May 2020, relevant data were collected completely anonymously from electronic medical files and recorded in an anonymized database (S1 Table).

The study was conducted in accordance with the Declaration of Helsinki. The ethics committee of the Hamburg Chamber of Physicians, to which we requested authorization for the project, determined that local legislation exempts this retrospective analysis of anonymized data derived from routine care from the need for specific ethical approval and informed consent (file reference PV7252). The STROBE guidelines (Strengthening Reporting on Observational Studies in Epidemiology) were employed to review reporting in this study [12].

## Preoperative assessment

The medical history was recorded with all relevant demographic and medical parameters. Assessment and determination of indication for operation was performed by duplex ultrasonography in the standing position using a Logiq P6 Pro (GE Healthcare, Chicago, IL, USA). A treatment was offered to the patient for venous insufficiency in Clinical, Etiological, Anatomy and Pathophysiology (CEAP) Class C2-C6 in combination with duplex sonographic reflux duration > 1 sec in the groin level of the diseased GSV. The anatomy of GSV was further examined for diameter, presence and diameter of ascending tributaries and their distance from the SFJ (Fig 1A). For this purpose, the corresponding veins were carefully examined and measured in different sectional planes with the linear transducer.

The strategy of ablation was established and documented during the preoperative examination. The aim was to switch off the GSV and all its ascending tributaries as far as technically feasible. An additional treatment of a tributary was planned if it appeared to be of sufficient caliber for cannulation and joined the GSV close to the SFJ. In contrast, additional ablation of the lateral branches was avoided if the distance to the SFJ was assumed to be sufficient and the diameter small enough. In such cases we tried to achieve sealing by supplying adequate thermal energy to the ostium of the tributaries.

## Procedures

The operations were carried out either under general anesthesia with tumescence using saline solution, or under tumescent local anesthesia (1000 mL physiological saline + 50 mL Mepivacaine 1% + 8 mL sodium bicarbonate 8.4%) alone. Ultrasonography with a portable ultrasound system (Logiq e, GE Healthcare, Chicago, IL, USA) was utilized to identify the GSV. The 1470nm 2-ring radial fibers (Biolitec AG, Jena, Germany) were used, either the 6 Fr fiber with 10 W for larger diameters, or the 4 Fr Slim-fiber at 8 W. The GSV was cannulated below or above the knee. In cases in which simultaneous endovenous ablation of the ascending tributaries was indicated on the basis of the preoperative examination, these were already punctured under ultrasound guidance and secured with a guide wire or an indwelling vein cannula, depending on the fiber diameter used. Thereafter, the laser fiber was inserted in the GSV under ultrasound guidance, the tip of the fiber being exactly at the SFJ (Fig 1B). After infusion of tumescence solution, the catheter position was checked again (Fig 1C). At the SFJ, a full cycle of 80–130 J, dependent on vein diameter, was applied without pullback. Thereafter the fiber was pulled 0.5 cm and another cycle of 80–130 J applied (Fig 1D). This was followed by continuous withdrawal. Energy transfer was reduced to 60–80 J/cm at distal parts of the thigh. After finishing the treatment of the GSV, the tributary was then disabled, if necessary. When using the 6 Fr fiber, the venous sheath was inserted beforehand via the already inserted guide wire according to Seldinger. Endothermal ablation of the tributaries was then carried out at 60–80 J/cm. Apart from the EVLA procedure, no adjunctive treatment techniques, such as

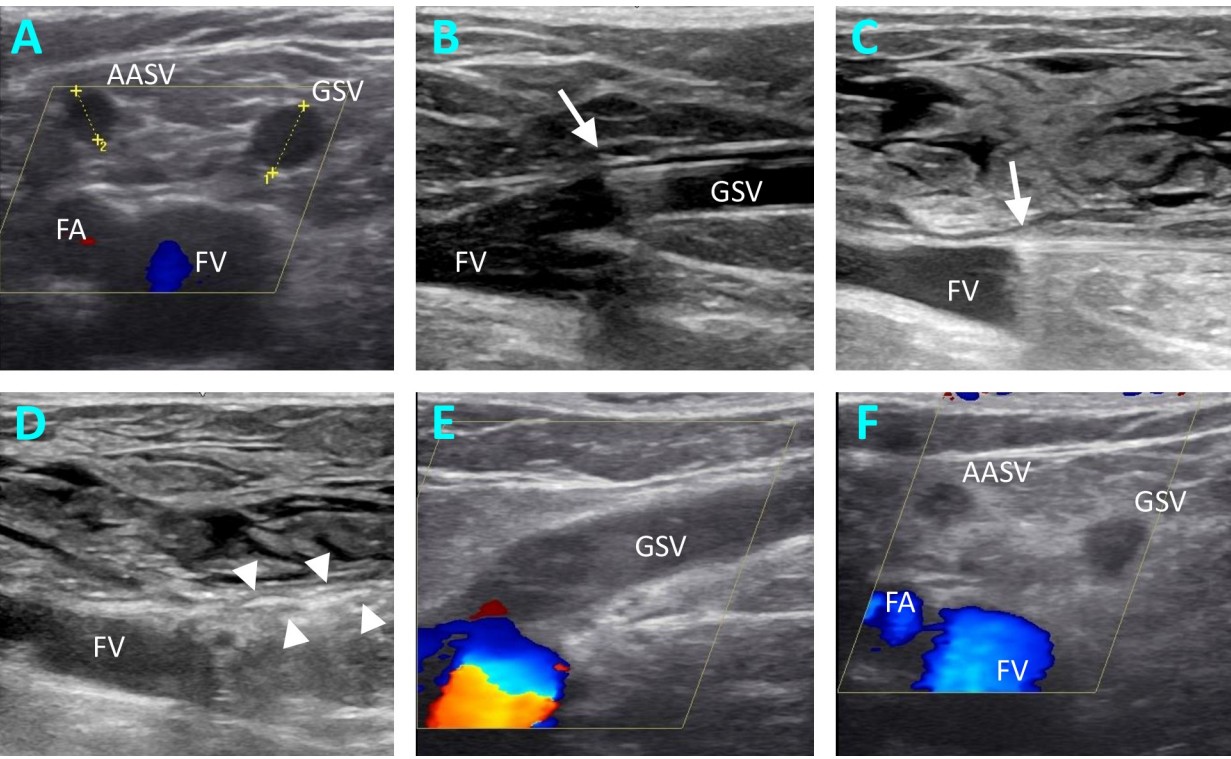

**Fig 1. EVLA treatment of superficial venous reflux of GSV.** Ultrasound images pre-, intra- and postoperative. A, measuring the diameters of GSV and AASV in transverse section in right-sided GSV insufficiency. B, intraoperative ultrasound image, where the placement of the laser fiber (arrow) is documented. C, intraoperative ultrasound image to document the catheter tip position (arrow) after infusion of the peritumescent solution. D, intraoperative ultrasound image showing the thermal reaction (arrow heads) in the GSV immediately after laser ablation. E, F, Ultrasound findings at the follow-up examination, showing the flush occlusion of the GSV in longitudinal section (E) and the ablation success of the GSV and AASV in cross-section (F). AASV, anterior accessory saphenous vein; FA, femoral artery; FV, femoral vein; GSV, great saphenous vein.

phlebectomies or sclerotherapy, were used in the proximal GSV or the GSV tributaries in the groin area.

Treatment-related parameters, like success of puncture and lengths of ablated vein segments were documented. Low-molecular heparin as thrombosis prophylaxis was given intraoperatively, and extended for 3 to 5 days postoperatively in presence of pro-thrombotic risk factors. Compression stockings after surgery have generally not been used. Patients were advised to schedule the follow-up examination for the 10th postoperative day. Both clinical and duplex sonographic controls were performed. The success of treatment and the patency of the deep vein system were documented.

### Primary and secondary endpoints

Early technical success as determined by the postoperative control examination was defined as the primary endpoint. This refers to the flush occlusion of the GSV including any present AASV or PASV (Fig 1E and 1F). For the GSV, this means that it is occluded along the entire treated length up to the SFJ, whereby an open superficial epigastric vein or an open proximal segment < 0.5 cm was not considered a failure. For the tributaries this meant that their confluence with the GSV is clearly thermally closed, or that their proximal segment is occluded over at least 2 cm (Fig 1F). The secondary endpoint was the frequency of endothermal heat-induced thrombosis (EHIT).

## Statistics

For the statistical analyses GraphPad Prism, version 8.4.2. was applied (GraphPad Software Inc, San Diego, CA, USA). Continuous data, such as age, body-mass index (BMI), and the diameters of GSV and tributaries were summarized by mean value and standard deviation (SD). Categorical variables, including baseline parameters, anatomical and treatment-associated variables, were reported as frequencies and percentages. Univariate analyses of the distribution of dichotomous categorical data were performed by using the Fisher's Exact Test. These dichotomous variables were gender, anatomical data such as side, proximity of the tributary to the SFJ, side of the tributary, and treatment setting associated data such as anesthesia, fiber type, and concomitant treatment of the tributary. Continuous data were evaluated using the Mann-Whitney U test, analyzing age, BMI, GSV and tributary diameters.

Multiple logistic regression using stepwise backward elimination was then used to further describe the relationship between technical success and the potentially predicting variables. We primarily included the parameters age, gender, side, diameter of GSV and tributaries, confluence of the tributary close to the SFJ, anterior tributary, intravenous anesthesia, fiber type and concomitant treatment of the tributary as independent variables. The stepwise elimination of the parameters was performed with simultaneous calculation of the Akaike Information criterion (AIC) with the purpose that the final model is given the best possible goodness of fit of the model. In the multiple logistic regression, the odds ratios (OR) and the 95% confidence intervals as well as the significance were given. For all analyses, a two-tailed p-value $< 0.05$ is assumed to be statistically significant.

# Results

From May to December 2019, 278 limbs were treated in 213 patients. Only one limb was treated in 148 cases (69.5%). In 48 patients (22.5%), both legs were treated in one session. In 17 patients (8%), both legs were treated in separate sessions. The mean age of the patients was 52.1 (15.6) years. Seventy-five patients (35.2%) were male, 138 (64.8%) of patients were female. The mean body mass index was 25.4 (5.0) kg/m$^2$. A CEAP stage of C2 was present in 157 limbs (56.5%). 94 limbs (33.8%) had stage C3, 25 cases (9.0%) stage C4. A classification as C6 was given in 2 cases (0.7%).

## Frequency and types of existing tributaries in the SFJ area

An assignment was made during the preoperative duplex examination based on the anatomical pattern of the SFJ in connection with its ascending tributaries (Fig 2A). First, we documented cases in which no recognizable tributary flows into the GSV within 3 cm of the SFJ. This favorable constellation from the operator's point of view was given in n = 28 cases (10.1%) and meant that flush ablation of the GSV is usually sufficient to achieve technical success. Then there have been cases in which an ascending side branch joins the GSV within the last 3 cm of the length of the GSV, but at a distance of more than 1 cm from the SFJ. This was the case in 70 limbs (25.2%), but here too the condition is favorable. By placing the catheter tip up to the SFJ and transmitting sufficient laser energy, the ostium of the AASV or PASV would also be thermally sealed with a high probability. In the remaining cases we saw a connection of the tributary, which was very close to the SFJ. Either into the GSV, at a distance of up to 1 cm, or directly into the SFJ, or after joining with descending, superficial iliac, epigastric or pudendal veins via a common trunk. This situation occurred in 180 limbs (64.7%). Since in this case even laser ablation of the GSV as a flush occlusion does not always provide sufficient sealing, we aimed for an additional ablation of these tributaries by separate cannulation (Fig 2A).

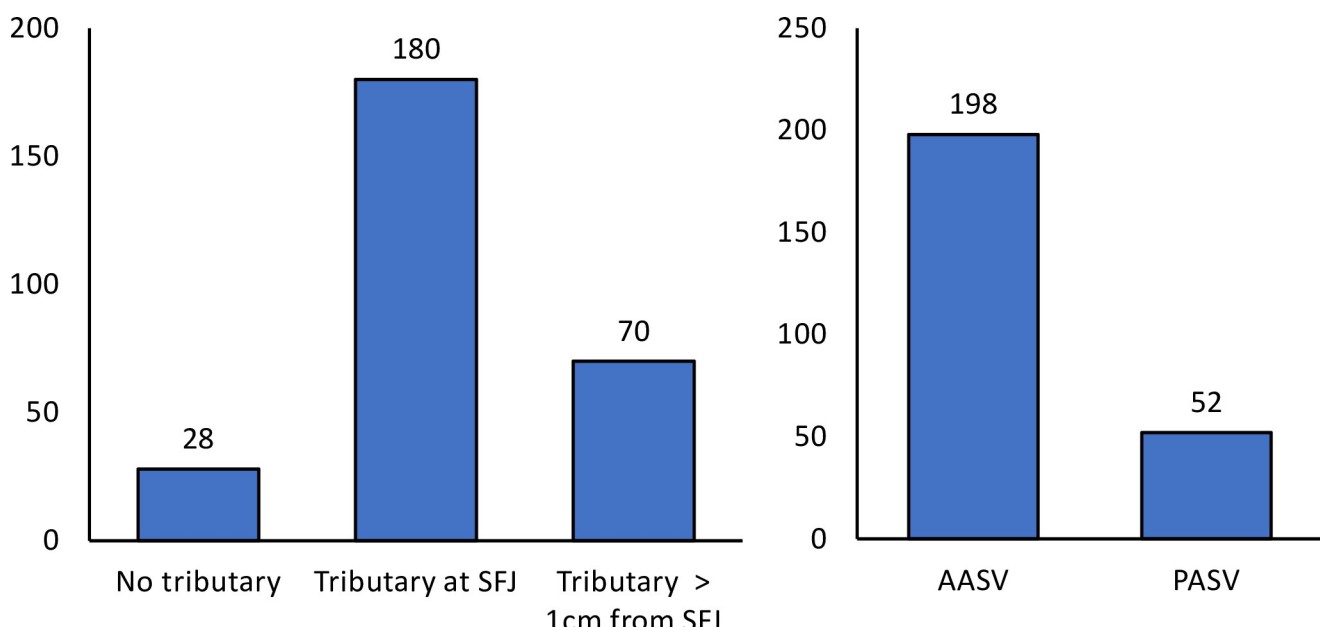

**Fig 2. Bar chart illustrating the patterns in which the uppermost tributaries join the GSV/SFJ, with an indication of the frequencies.** A, distinction in terms of distance from the SFJ. B, differentiation in terms of type. AASV, anterior accessory saphenous vein; PASV, posterior accessory saphenous vein, SFJ, sapheno-femoral junction.

In addition, the type of tributary, anterior versus posterior, was documented. In the event that both tributaries existed, the case was defined based on the uppermost branch. As further shown in Fig 2B, the AASV was encountered much more frequently than the PASV.

## Feasibility of the procedures and intraoperative outcome

On the basis of the preoperative ultrasound diagnostics, it was determined individually for each treatment whether an additional ablation of the tributaries would be performed in addition to the flush occlusion of the GSV. This was planned in 178 cases, but was not successful in 7 of these treatments (3.9%), the reason being failure to puncture or cannulate these veins. Hence, in 171 instances (61.5%) targeted laser ablations of the highest ascending tributaries were executed in addition to GSV ablation. In 4 out of these cases (2.3%), the ultrasonic morphological thermal reaction was determined not to be typical, so that a paravascular misplacement of the laser fiber could not be excluded. For subsequent analyses, however, these cases were kept in the group of treatments with concomitant laser ablation of a tributary.

## Initial technical success rate and complications

The overall, initial technical success was assessed on the basis of the postoperative duplex examination. The 10th day after treatment was recommended to the patients as the time of examination. The follow-up rate was 100%, and the mean duration to the follow up was 10.7 (4.5) days. Sufficient occlusion of the GSV was found in 277/278 (99.6%) of the treated limbs. Sufficient closure of ascending SFJ tributaries, if any, was achieved in 231 of 250 such cases (92.4%). Taken together, this resulted in 258 of the 278 cases (92.8%) being found to be technically successful.

EHIT grade 2 occurred in 2 patients (0.7%) and EHIT grade 3 in 1 patient (0.4%). In all cases, anticoagulation was administered in therapeutic doses and the EHIT completely

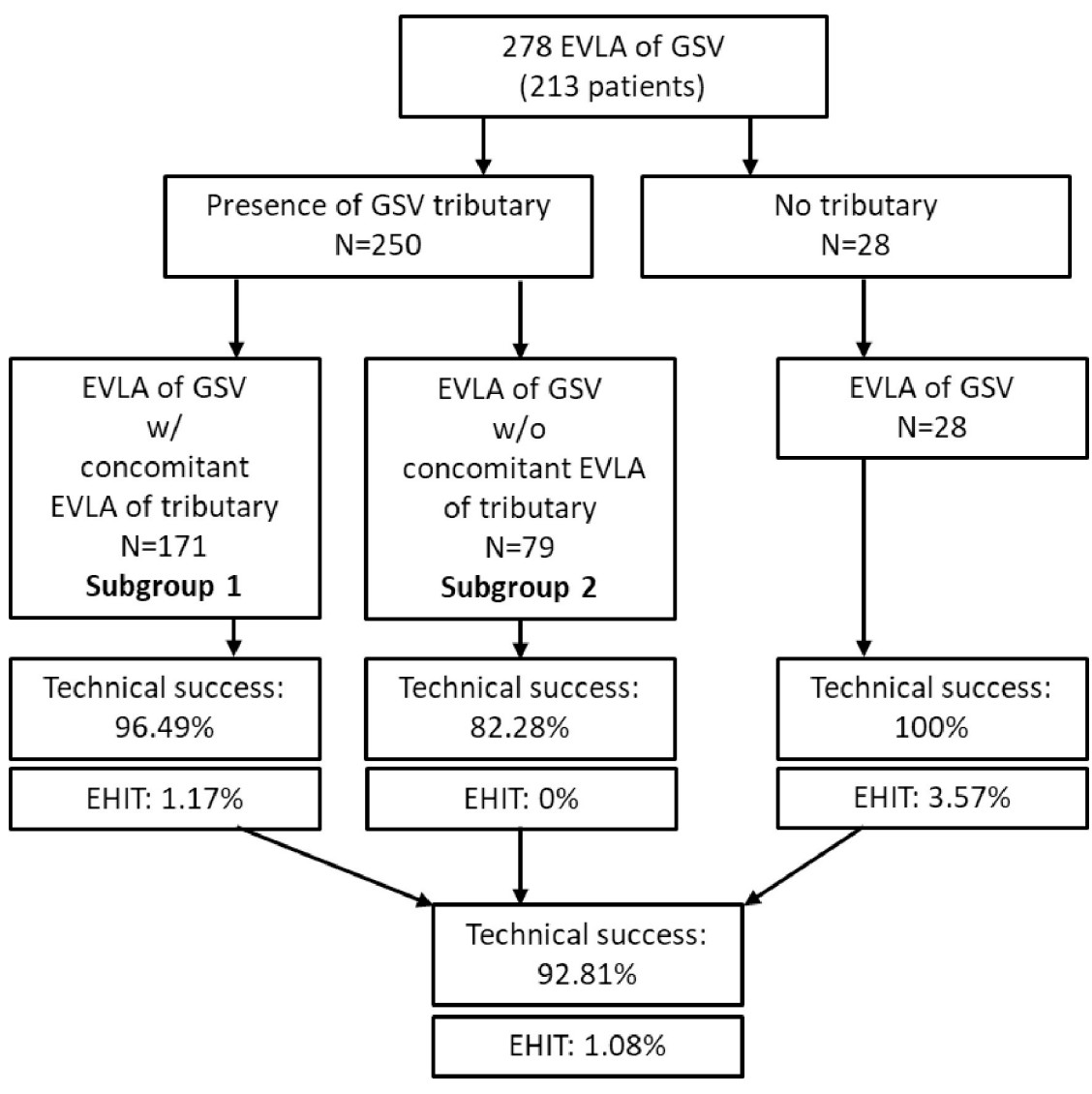

**Fig 3. Flow chart of the study.**

disappeared. There were no major complications and no cases of thromboembolism or nerve injury.

All 28 cases in which no tributary joining within 3 cm of the SFJ proved to be technical successes (Fig 3). The 250 treatments where a tributary was identified in the SFJ area were analyzed in detail with reference to their anatomy. For clarification, a flow chart of the analysis is shown in Fig 3.

## Characteristics of cases with and without concomitant ablation of a tributary

The following Table 1 shows the characteristics of the 250 cases in which such a tributary occurred in the SFJ area. Here the cases with (subgroup 1, n = 171) and without additional ablation (subgroup 2, n = 79) of the tributaries are contrasted. The subgroup 2 comprises cases, whereby additional ablation was either not planned due to preoperative ultrasound diagnostics (n = 72) or the puncture or cannulation did not succeed (n = 7).

**Table 1. Baseline, anatomical and treatment characteristics of cases with (subgroup 1) and without (subgroup 2) concomitant laser ablation of a GSV tributary.**

| Parameter | Subgroup 1 | Subgroup 2 | p-value |
|---|---|---|---|
| | w/concomitant EVLA of GSV tributary | w/o concomitant EVLA of GSV tributary | |
| N treated limbs | 171 | 79 | |
| Age, years, mean (SD) | 49.76 (14.85) | 50.78 (15.08) | 0.582 |
| Female, n (%) | 104 (60.82) | 58 (73.42) | 0.064 |
| BMI, kg/m$^2$, mean (SD) | 25.81[a] (5.56) | 25.54[b] (4.71) | 0.503 |
| GSV diameter, mm, mean (SD) | 7.63 (2.17) | 7.86 (1.83) | 0.162 |
| Tributary diameter, mm, mean (SD) | 4.15 (2.00) | 2.33 (0.98) | < 0.001 |
| SFJ anatomy | | | |
| Tributary joins at SFJ, n (%) | 142 (83.04) | 38 (48.10) | < 0.001 |
| Tributary joins GSV > 1cm from SFJ, n (%) | 29 (16.96) | 41 (51.90) | |
| AASV, n (%) | 133 (77.78) | 65 (82.28) | 0.503 |
| PASV, n (%) | 38 (22.22) | 14 (17.72) | |
| Use of intravenous sedative agents, n (%) | 122 (71.35) | 48 (60.76) | 0.109 |
| Size of laser fiber used | | | |
| 6 French, n (%) | 141 (82.46) | 60 (75.95) | 0.235 |
| 4 French, n (%) | 30 (17.54) | 19 (24.05) | |
| Technical success, n (%) | 165 (96.49) | 65 (82.28) | < 0.001 |
| GSV occluded, n (%) | 170 (99.42) | 79 (100) | 1.000 |
| Tributary occluded, n (%) | 166 (97.08) | 65 (82.28) | < 0.001 |
| EHIT Grade > I | 2 (1.17) | 0 (0) | 1.000 |

AASV, anterior accessory saphenous vein; BMI, body mass index; EHIT, endovenous heat-induced thrombosis; GSV, great saphenous vein. PASV, posterior accessory saphenous vein.

a, data missing in 13 cases

b, data missing in 3 cases.

The two subgroups did not differ significantly in terms of demographic parameters and the diameter of the GSV to be treated (Table 1). As expected from the treatment strategy stated above, the mean diameter of the tributaries was smaller in subgroup 2. Similarly, as a result of the treatment approach, the frequency of tributary entry in the immediate proximity of the SFJ (<1cm) was higher in subgroup 1 than in subgroup 2 (Table 1). These significant differences between the two groups result from the selection of cases based on preoperative examination.

Furthermore, this view indicated a significantly greater early technical success for the cases represented in subgroup 1 where additional ablation of the tributary was performed. Although there were two cases with an EHIT in subgroup 1 versus 0 in subgroup 2, this difference was not significant.

## Analysis of potential predicting factors for technical success

In the following we have investigated whether there were predicting factors that influence the rate of technical success. Initially, the 28 cases without a joining ascending tributary within 3 cm of the SFJ were technical successes according to our definition. However, these cases had to be excluded for the subsequent logistic regression analysis due to the otherwise existing problem of perfect separation.

We then focused the further analysis of influencing factors on those 250 cases where a tributary was present in the SFJ area. The stepwise, backward multiple logistic regression analysis initially included all parameters that were also compared in the subgroup analysis. Only the

**Table 2. Multiple logistic regression analysis of factors influencing the technical success.**

| Parameter | OR | 95% CI | p-value |
|---|---|---|---|
| Intercept | 1.186 | 0.2903 to 5.319 | 0.8152 |
| Tributary joins at SFJ | 0.2253 | 0.05456 to 0.7681 | 0.0253 |
| Anterior Tributary | 3.959 | 1.142 to 13.73 | 0.027 |
| Use of intravenous sedative agents | 2.335 | 0.8205 to 6.742 | 0.1103 |
| Use of 6 French laser fiber | 2.565 | 0.8397 to 7.624 | 0.0906 |
| Concomitant laser ablation of tributary | 10.39 | 3.420 to 36.15 | <0.0001 |

SFJ, saphenofemoral junction.

BMI was excluded, as in some cases no data were available. In the best fitting, final model, a significant positive prediction between the technical success and the additional ablation of the side branch is maintained (Table 2). The presence of an AASV, in contrast to the PASV, is also correlated with technical success, while the inflow of the tributary close to SFJ (<1cm) represents a risk factor (Table 2).

## Discussion

Our data suggest that the technical success of EVLA could be positively influenced by prophylactic ablation of the ascending tributaries, which enter the GSV in the SFJ area. And that there is sufficient feasibility and safety regarding the targeted ablation of these side branches. A current prospective study has already reported that a flush occlusion, in which the laser fiber is placed right up to the SFJ, was feasible and safe [8]. In this study, based on 135 treatments, an EHIT was detected in 1.6% of the treatments, which is comparable to the low frequency of 1.1% in our own sample, and also comparable to the reported frequencies of EHIT in the literature, with a safety distance of 1–3 cm between the catheter tip and the SFJ usually maintained here [13]. In a recent case series of 34 cases of recurrent varicose veins from the SFJ treated with flush EVLA, we observed low morbidity and no EHIT [14].

The co-treatment of varicose side branches during the primary treatment by foam sclerotherapy [15], mini-phlebectomy [16, 17] or even laser [18] is widely applied. To date, however, there is admittedly no clear scientific evidence that targeted thermal treatment of non-expanded or non-refluxing tributaries has any prognostic significance, although a benefit appears to be apparent given the long-term study data comparing high ligation and stripping and EVLA [4, 5]. Proponents of surgical high ligation argue, based on decades of experience, that a flush ligation should be performed, with removal and, if possible, distancing of the severed venous stumps in order to keep the risk of recurrence as low as possible [10, 19]. This is founded on the concept that after they have been transected, the lateral branches potentially reconnect and promote the formation of recurrent varicose veins. In contrast, stripping or avulsion of the detached distal venous segments may prevent from such reconnection. In our own center, in which more than 35000 thermal vein treatments have been performed since its foundation in 2005, we are therefore striving to achieve long-distance thermal closure of even non-extended side branches in the SFJ area.

However, there are also studies that support a contrary view [20–22]. According to these findings, less invasive dissection in the SFJ region, leaving competent tributaries in place, might have advantages, possibly due to less venous stasis, less local inflammation, and thus less neovascularization. Correspondingly, more radical laser ablation of competent tributaries may have the inherent potential to increase neovascularization, and this concern, not yet reflected in our own experience, needs to be clarified by subsequent studies.

As already described in the methods, there has not yet been a clearly defined criterion regarding the question of whether a tributary should also be occluded by additional cannulation and ablation. In the present sample, an additional occlusion was always sought if the confluence of the tributary was within 1 cm of the SFJ, or if there was a common trunk with cranial veins. The rationale here was that in the immediate vicinity of the SFJ, despite the placement of the laser-emitting element directly at the transition to the deep vein, the necessary heat energy cannot always be applied to ensure that the confluences of the tributaries are also closed. A recent Italian study with a mean observation period of 29.7 months seems to support the estimation that the distance of the inflowing tributaries from the SFJ has an influence on the recurrence rate [23]. A Cox regression analysis revealed that a direct inflow of the AASV into the SFJ is a risk factor for recurrence. However, comparability is limited by the fact that the radiofrequency method was used in this study, and a flush occlusion and targeted ablation of the tributaries was not attempted.

The present study has some limitations, which need to be discussed. First, the study design without short-term, mid-term or long-term observation does not allow conclusions from which prognostic relevance for preventive ablation of the tributaries can be deduced. A comparative study between elimination and preservation of the ascending tributaries and their influence on the recurrence rate would be of major interest here. Secondly, in some cases, a thrombotic, non-fibrotic occlusion may be present in the early follow-up, so that a deterioration of the results after longer observation cannot be excluded. We suspect that this is especially true in cases where no prophylactic ablation of the tributaries is performed, and whose orifices initially seem to be sealed by the flush occlusion. Thirdly, due to the large confidence intervals (Table 2) resulting from the limited number of events with failure in the present study (n = 20), the need for larger studies to assess the possible influence of predictors on technical success arises.

The strength of this study is that we collected evidence in an observational setting that performing EVLA as a flush occlusion does not automatically abolish the upmost GSV tributaries sufficiently. Successful simultaneous ablation during flush EVLA increases the rate of closure of these veins. Following this, prospective studies could further establish whether there is a reasonable maximum distance from the SFJ or a minimum diameter at which closure of an inflowing tributary by additional cannulation and ablation is recommended.

## Conclusions

According to the present case series, concomitant laser ablation of the ascending tributaries of the GSV appears feasible and increases the probability of technical success when their closure is desired, particularly when the confluence is very close to the SFJ. The differentiation and adaptation of the ablation strategy to anatomic patterns of the SFJ and the joining tributaries may be further established by future prospective studies. Whether prophylactic occlusion of non-refluxing tributaries prevent recurrence formation in the long-term also needs to be further investigated.

## Supporting information

**S1 Table. Raw data.**
(XLSX)

## Acknowledgments

The authors would like to thank Dr. Kai Brüssau, Institute of Information Systems, University of Hamburg, for his statistical support.

## Author Contributions

**Conceptualization:** Lars Müller.

**Data curation:** Lars Müller.

**Formal analysis:** Lars Müller.

**Investigation:** Lars Müller, Jens Alm.

**Methodology:** Lars Müller.

**Project administration:** Lars Müller, Jens Alm.

**Resources:** Lars Müller.

**Software:** Lars Müller.

**Supervision:** Lars Müller, Jens Alm.

**Validation:** Lars Müller.

**Visualization:** Lars Müller.

**Writing – original draft:** Lars Müller.

**Writing – review & editing:** Lars Müller, Jens Alm.

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
