## [Decision Letter · Decision Letter 0]

8 Dec 2020

PONE-D-20-35887

Feasibility and potential significance of prophylactic ablation of the major ascending tributaries in endovenous laser ablation of the great saphenous vein: A case series

PLOS ONE

Dear Dr. Müller,

Thank you for submitting your manuscript to PLOS ONE. After careful consideration, we feel that it has merit but does not fully meet PLOS ONE’s publication criteria as it currently stands. Therefore, we invite you to submit a revised version of the manuscript that addresses the points raised during the review process.

The reviewers have commented on your above paper. They have suggested that this manuscript be revised according to the reviewers suggestions and resubmitted.  Provided you address the changes recommended, the manuscript will be accepted for publication. 

We look forward to receiving your revised manuscript.

Kind regards,

Prof. Raffaele Serra, M.D., Ph.D

Academic Editor

PLOS ONE

Additional Editor Comments:

The reviewers have commented on your above paper. They have suggested that this manuscript be revised according to the reviewers suggestions and resubmitted.

Journal Requirements:

2.  In the ethics statement in the manuscript and in the online submission form, please provide additional information about the patient records/samples used in your retrospective study, including: a) whether all data were fully anonymized before you accessed them; b) the date range (month and year) during which patients' medical records/samples were accessed.

We note that one or more of the authors are employed by a commercial company: Dermatologikum Hamburg GmbH.

3.1. Please provide an amended Funding Statement declaring this commercial affiliation, as well as a statement regarding the Role of Funders in your study. If the funding organization did not play a role in the study design, data collection and analysis, decision to publish, or preparation of the manuscript and only provided financial support in the form of authors' salaries and/or research materials, please review your statements relating to the author contributions, and ensure you have specifically and accurately indicated the role(s) that these authors had in your study. You can update author roles in the Author Contributions section of the online submission form.

3.2. Please also provide an updated Competing Interests Statement declaring this commercial affiliation along with any other relevant declarations relating to employment, consultancy, patents, products in development, or marketed products, etc.  

Reviewers' comments:

Reviewer's Responses to Questions

**Comments to the Author**

1. Is the manuscript technically sound, and do the data support the conclusions?

Reviewer #1: Yes

2. Has the statistical analysis been performed appropriately and rigorously? 

Reviewer #1: Yes

3. Have the authors made all data underlying the findings in their manuscript fully available?

Reviewer #1: Yes

4. Is the manuscript presented in an intelligible fashion and written in standard English?

Reviewer #1: Yes

5. Review Comments to the Author

Reviewer #1: A. Overall evaluation and general comments

1.The study is designed in such a way that cases of EVLA w/o big untreated tributaries are missing. Thus no effective comparison can be performed and no conclusion can be drawn about the role of tributary EVLA in the SFJ.

2. The development of varicose veins from previously non-enlarged ascending tributaries of the GSV can also occur as a compensatory reaction to the reduction of the venous patrimony due to the ablation strategy.

Suppressing the tributaries, especially when competent ones, could aggravate the dilatation of the remaining veins of the thigh.

Consider this possibility in the discussion.

B. Detailed evaluation of specific deficiencies with suggestions for improvements

1.300 Table 1

You used a ultrasound criterion to subdivide the patients into the 2 groups, thermal ablation with and without tributary avulsion.

Cases were assigned to the 2 groups according to the caliber ot the tributaries and to the distance from the SFJ. Thus the significant difference of the increased tributary diameter and the small distance from the SFJ could is the effect of the selection of cases.

2.155-156 > The endothermal ablation of tributaries was executed thereafter at 60-80 J/cm.,

It's necessary to add more details on the tributary treatment. Did you perform a separate cannulation ? Did you cannulate them before or after the tumescence ? Tumescence generally doesn't allow a clear visualization, thus these details are important.

6. PLOS authors have the option to publish the peer review history of their article (what does this mean?). If published, this will include your full peer review and any attached files.

Reviewer #1: No

---

## [Author Response · Author response to Decision Letter 0]

18 Dec 2020

Dear Professor Serra, 

I would like to thank, also on behalf of my co-author Dr. Jens Alm, for the helpful review process. I also would like to thank the reviewers for their dedicated time and effort with our article.

We have revised our manuscript in several respects on the basis of your criticism and suggestions and would now like to explain them to you. For this purpose, we will precede the comments (in bold letters) made by you and by the reviewer and then display our explanations and the respective changes in the manuscript:

We have checked and revised the names of the image files.

2. In the ethics statement in the manuscript and in the online submission form, please provide additional information about the patient records/samples used in your retrospective study, including: a) whether all data were fully anonymized before you accessed them; b) the date range (month and year) during which patients' medical records/samples were accessed.

We have revised this relevant section of text (lines 121-128) as well as the online submission form.

3. Thank you for stating the following in the Competing Interests section: "The authors have declared that no competing interests exist." We note that one or more of the authors are employed by a commercial company: Dermatologikum Hamburg GmbH.

3.1. Please provide an amended Funding Statement declaring this commercial affiliation, as well as a statement regarding the Role of Funders in your study. If the funding organization did not play a role in the study design, data collection and analysis, decision to publish, or preparation of the manuscript and only provided financial support in the form of authors' salaries and/or research materials, please review your statements relating to the author contributions, and ensure you have specifically and accurately indicated the role(s) that these authors had in your study. You can update author roles in the Author Contributions section of the online submission form.

The funding statement should be updated as follows: “Both authors of this study are employed by Dermatologikum Hamburg GmbH. As a non-academic, commercial company, the employer and funder provided support in the form of salaries for authors [L.M., J.A.], but did not have any additional role in the study design, data collection and analysis, decision to publish, or preparation of the manuscript. The specific roles of these authors are articulated in the ‘author contributions’ section.”

3.2. Please also provide an updated Competing Interests Statement declaring this commercial affiliation along with any other relevant declarations relating to employment, consultancy, patents, products in development, or marketed products, etc. 

The Competing Interests Statement should be updated as follows: “As authors of this study [L.M., J.A.], we declare that we are employed by Dermatologikum Hamburg GmbH, a nonacademic, commercial company. This does not alter our adherence to PLOS ONE policies on sharing data and materials.”

Reviewer #1: 

A. Overall evaluation and general comments

1. The study is designed in such a way that cases of EVLA w/o big untreated tributaries are missing. Thus no effective comparison can be performed and no conclusion can be drawn about the role of tributary EVLA in the SFJ.

We would like to express, of course, that the focus is on the realistic technical possibility of ablating tributaries, but without knowing how this will affect the long-term prognosis. To emphasize this even more, we have added the last sentence ("Its effect on the recurrence rate needs further research") in the abstract, in addition to minor text changes. 

In addition, we have added a similar statement in the Discussion (lines 377-388: "First, the study design without short-term, mid-term, or long-term observation does not allow conclusions from which prognostic relevance for preventive ablation of the tributaries can be deduced. A comparative study between elimination and preservation of the ascending tributaries and their influence on the recurrence rate would be of major interest here.")

2. The development of varicose veins from previously non-enlarged ascending tributaries of the GSV can also occur as a compensatory reaction to the reduction of the venous patrimony due to the ablation strategy. Suppressing the tributaries, especially when competent ones, could aggravate the dilatation of the remaining veins of the thigh. Consider this possibility in the discussion.

We are aware that there are indeed study data that can be used to question the tactics of ablation described here. We would like to bring this valuable aspect into the discussion by providing additional literature in this regard (refs. 20-22) and a corresponding paragraph. (Line 556-361: ”However, there are also studies that support a contrary view [20-22]. According to these findings, less invasive dissection in the SFJ region, leaving competent tributaries in place, might have advantages, possibly due to less venous stasis, less local inflammation, and thus less neovascularization. Correspondingly, more radical laser ablation of competent tributaries may have the inherent potential to increase neovascularization, and this concern, not yet reflected in our own experience, needs to be clarified by subsequent studies.”)

B. Detailed evaluation of specific deficiencies with suggestions for improvements

1. 300 Table 1 You used a ultrasound criterion to subdivide the patients into the 2 groups, thermal ablation with and without tributary avulsion. Cases were assigned to the 2 groups according to the caliber ot the tributaries and to the distance from the SFJ. Thus the significant difference of the increased tributary diameter and the small distance from the SFJ could is the effect of the selection of cases.

Quite rightly, we are dealing here with a weighting that results from the indication to one or another ablation tactic. This is ultimately also the aim of this presentation, and of the underlying text, from lines 292-302. However, we have added another sentence to draw attention to this. (line 301-302: "These significant differences between the two groups result from the selection of cases based on preoperative examination. ")

2.155-156 > The endothermal ablation of tributaries was executed thereafter at 60-80 J/cm.,

It's necessary to add more details on the tributary treatment. Did you perform a separate cannulation ? Did you cannulate them before or after the tumescence ? Tumescence generally doesn't allow a clear visualization, thus these details are important.

We have tried to expand the corresponding text section substantially in the hope that our technical procedure will now be better understood (line 154-157 and line 162-165).

Other changes:

-We have already defined the abbreviation EVLA in the title (line 4).

-In the abstract we have revised the text a bit more without changing the content.

-Some wording we have also changed in the introduction, especially in the first 3 sentences.

-In the discussion we removed one passage (lines 368-371), this passage partly coincided with contents already presented above.

I look forward to hearing from you regarding our submission. We will be happy to continue to answer any further suggestions or queries or to make any necessary changes.

Yours sincerely, 

Dr. Lars Müller

---

## [Editor Report · Decision Letter 1]

28 Dec 2020

Feasibility and potential significance of prophylactic ablation of the major ascending tributaries in endovenous laser ablation (EVLA) of the great saphenous vein: A case series

PONE-D-20-35887R1

Dear Dr. Müller,

We’re pleased to inform you that your manuscript has been judged scientifically suitable for publication and will be formally accepted for publication once it meets all outstanding technical requirements.

Kind regards,

Prof. Raffaele Serra, M.D., Ph.D

Academic Editor

PLOS ONE

Additional Editor Comments (optional):

amended manuscript is acceptable
---

## [Editor Report · Acceptance letter]

30 Dec 2020

PONE-D-20-35887R1 

Feasibility and potential significance of prophylactic ablation of the major ascending tributaries in endovenous laser ablation (EVLA) of the great saphenous vein: A case series 

Dear Dr. Müller:

I'm pleased to inform you that your manuscript has been deemed suitable for publication in PLOS ONE. Congratulations! Your manuscript is now with our production department. 

Kind regards, 

on behalf of

Prof. Raffaele Serra 

Academic Editor

PLOS ONE